# Meta-Analysis of Mutations in *ALOX12B* or *ALOXE3* Identified in a Large Cohort of 224 Patients

**DOI:** 10.3390/genes12010080

**Published:** 2021-01-09

**Authors:** Alrun Hotz, Julia Kopp, Emmanuelle Bourrat, Vinzenz Oji, Katalin Komlosi, Kathrin Giehl, Bakar Bouadjar, Anette Bygum, Iliana Tantcheva-Poor, Maritta Hellström Pigg, Cristina Has, Zhou Yang, Alan D. Irvine, Regina C. Betz, Giovanna Zambruno, Gianluca Tadini, Kira Süßmuth, Robert Gruber, Matthias Schmuth, Juliette Mazereeuw-Hautier, Natalie Jonca, Sophie Guez, Michela Brena, Angela Hernandez-Martin, Peter van den Akker, Maria C. Bolling, Katariina Hannula-Jouppi, Andreas D. Zimmer, Svenja Alter, Anders Vahlquist, Judith Fischer

**Affiliations:** 1Institute of Human Genetics, Medical Center, Faculty of Medicine, University of Freiburg, 79106 Freiburg, Germany; alrun.hotz@uniklinik-freiburg.de (A.H.); julia.kopp@uniklinik-freiburg.de (J.K.); katalin.komlosi@uniklinik-freiburg.de (K.K.); maggie_2004_2008@126.com (Z.Y.); andreas.zimmer@uniklinik-freiburg.de (A.D.Z.); svenja.alter@uniklinik-freiburg.de (S.A.); 2Department of Dermatology, Reference Center for Rare Skin Diseases MAGEC, Saint Louis Hospital AP-HP, 75010 Paris, France; emmanuelle.bourrat@aphp.fr; 3Department of Dermatology and Venereology, Münster University Medical Center, 48149 Münster, Germany; vinzenz.oji@ukmuenster.de (V.O.); Kira.Suessmuth@ukmuenster.de (K.S.); 4Department of Dermatology and Allergy, University of Munich LMU, 80337 Munich, Germany; Kathrin.Giehl@med.uni-muenchen.de; 5Department of Dermatology, CHU of Bab-El-Oued Algiers, Algiers 16008, Algeria; bouadjar@yahoo.fr; 6Department of Dermatology, Odense University Hospital, 5000 Odense, Denmark; Anette.Bygum@rsyd.dk; 7Department of Clinical Genetics, Odense University Hospital, 5000 Odense, Denmark; 8Clinical Institute, University of Southern Denmark, 5000 Odense, Denmark; 9Department of Dermatology, University of Cologne, 50937 Cologne, Germany; iliana.tantcheva-poor@uk-koeln.de; 10Department of Clinical Genetics, Karolinska University Hospital, 17176 Stockholm, Sweden; maritta.hellstrom-pigg@sll.se; 11Department of Dermatology, Medical Center-University of Freiburg, University of Freiburg, 79104 Freiburg, Germany; cristina.has@uniklinik-freiburg.de; 12Department of Dermatology, Beijing Children’s Hospital, Capital Medical University, National Center for Children’s Health, Beijing 100045, China; 13Dermatology, Children’s Health Ireland and Clinical Medicine, Trinity College Dublin, D12 N512 Dublin, Ireland; irvinea@tcd.ie; 14Institute of Human Genetics, School of Medicine & University Hospital Bonn, University of Bonn, 53127 Bonn, Germany; regina.betz@uni-bonn.de; 15Genodermatosis Unit, Genetics and Rare Diseases Research Division, Bambino Gesù Children’s Hospital, IRCCS, 00165 Rome, Italy; giovanna.zambruno@gmail.com; 16Fondazione IRCCS Cà Granda Ospedale Maggiore Policlinico, UOSD Pediatria ad Alta Intensità di Cura, 20122 Milan, Italy; gtadinicmce@unimi.it (G.T.); sophie.guez@policlinico.mi.it (S.G.); michela.brena@policlinico.mi.it (M.B.); 17Department of Dermatology, Venereology and Allergy, Medical University of Innsbruck, 6020 Innsbruck, Austria; robert.gruber@tirol-kliniken.at (R.G.); Matthias.Schmuth@i-med.ac.at (M.S.); 18Reference Center for Rare Skin Diseases, Dermatology Department, CHU Larrey, Université Paul Sabatier, 31000 Toulouse, France; mazereeuw-hautier.j@chu-toulouse.fr; 19Department of Epidermis Differentiation and Rheumatoid Autoimmunity, UMR 1056 Inserm University Toulouse, Place du Dr Baylac, Hôpital Purpan, 31059 Toulouse, France; nathalie.jonca@udear.cnrs.fr; 20Department of Dermatology, Hospital Infantil Niño Jesús, 28009 Madrid, Spain; ahernandez@aedv.es; 21Department of Genetics, University Medical Center Groningen, University of Groningen, 9700RB Groningen, The Netherlands; p.c.van.den.akker@umcg.nl; 22Center for Blistering Diseases, Department of Dermatology, University Medical Center Groningen, University of Groningen, 9700RB Groningen, The Netherlands; m.c.bolling@umcg.nl; 23ERN-Skin Center, Department of Dermatology and Allergology, University of Helsinki and Helsinki University Central Hospital, 00029 HUS Helsinki, Finland; Katariina.Hannula-Jouppi@hus.fi; 24Folkhälsan Research Center, Helsinki, Finland and Research Programs Unit, Stem Cells and Metabolism Research Program, University of Helsinki, 00290 Helsinki, Finland; 25Department of Medical Sciences/Dermatology, Uppsala University, SE-751 85 Uppsala, Sweden; anders.vahlquist@medsci.uu.se

**Keywords:** ALOX12B, ALOXE3, ARCI, ichthyosis

## Abstract

The autosomal recessive congenital ichthyoses (ARCI) are a nonsyndromic group of cornification disorders that includes lamellar ichthyosis, congenital ichthyosiform erythroderma, and harlequin ichthyosis. To date mutations in ten genes have been identified to cause ARCI: *TGM1*, *ALOX12B*, *ALOXE3*, *NIPAL4*, *CYP4F22*, *ABCA12*, *PNPLA1*, *CERS3*, *SDR9C7*, and *SULT2B1*. The main focus of this report is the mutational spectrum of the genes *ALOX12B* and *ALOXE3*, which encode the epidermal lipoxygenases arachidonate 12-lipoxygenase, i.e., 12R type (12R-LOX), and the epidermis-type lipoxygenase-3 (eLOX3), respectively. Deficiency of 12R-LOX and eLOX3 disrupts the epidermal barrier function and leads to an abnormal epidermal differentiation. The type and the position of the mutations may influence the ARCI phenotype; most patients present with a mild erythrodermic ichthyosis, and only few individuals show severe erythroderma. To date, 88 pathogenic mutations in *ALOX12B* and 27 pathogenic mutations in *ALOXE3* have been reported in the literature. Here, we presented a large cohort of 224 genetically characterized ARCI patients who carried mutations in these genes. We added 74 novel mutations in *ALOX12B* and 25 novel mutations in *ALOXE3*. We investigated the spectrum of mutations in *ALOX12B* and *ALOXE3* in our cohort and additionally in the published mutations, the distribution of these mutations within the gene and gene domains, and potential hotspots and recurrent mutations.

## 1. Introduction

Autosomal recessive congenital ichthyosis (ARCI) comprises a clinically and genetically heterogeneous group of rare disorders of cornification characterized by hyperkeratosis, scaling of the body, and a variable degree of erythroderma. ARCI is subclassified into lamellar ichthyosis (LI), congenital ichthyosiform erythroderma (CIE), and harlequin ichthyosis (HI). Patients with LI often present with large, dark plate-like scales and with minimal erythema, whereas patients with CIE usually show variable erythroderma and generalized fine white scaling. Some patients show overlapping phenotypes of LI and CIE. HI represents the most severe form of ARCI and is a potentially life-threatening condition. CIE is typically on the milder end of the spectrum. Neonates with ARCI are often born with a collodion membrane, a parchment-like membrane covering the whole body surface. This membrane is shed within 1–3 weeks and is followed by presentation of an underlying LI or CIE. In some cases, either no or only mild signs of ichthyosis persist after the neonatal or infant period. This condition has been named self-improving collodion ichthyosis (SICI) and is often associated with mutations in *ALOX12B* and *ALOXE3* [1].

To date, mutations in ten genes have been identified to cause ARCI: *TGM1* (MIM 190195) [2], *ALOX12B* (MIM 603741) [3], *ALOXE3* (MIM 607206) [3], *NIPAL4*/*ICHTHYIN* (MIM 609383) [4], *CYP4F22* (MIM 611495) [5], *ABCA12* (MIM 607800) [6], *PNPLA1* (MIM 612121) [7], *CERS3* (MIM 615276) [8], *SDR9C7* (MIM 609769) [9], and *SULT2B1* (MIM 604125) [10]. Mutations in *TGM1* are the most common cause for ARCI, followed by mutations in the two lipoxygenase genes, *ALOX12B* and *ALOXE3* [11]. In this report we focus on the spectrum, the type, and the position of mutations in both *ALOX12B* and *ALOXE3* genes.

ARCI is associated with impaired skin barrier function, which is mainly caused by the inability of mutated keratinocytes to produce or to secrete the skin lipids that are required for the formation of the cornified cell envelope and for the extracellular lipid layers in the stratum corneum [12]. *ALOX12B* and *ALOXE3* encode the epidermal lipoxygenases arachidonate 12-lipoxygenase, i.e., 12R type (12R-LOX), and the epidermis-type lipoxygenase-3 (eLOX3), respectively. Lipoxygenases (LOXs) are a family of nonheme, iron-containing dioxygenases. LOXs catalyze the oxygenation of polyunsaturated fatty acid substrates that contain (Z,Z)-1,4-pentiadiene structures [13]. Deficiency of 12R-LOX and eLOX3 disrupts the epidermal barrier function and leads to an abnormal skin development.

The ALOX12B and ALOXE3 proteins have a similar structure that consists of two main domains: (a) the PLAT (Polycystin-1, Lipoxygenase, α-Toxin) or the LH2 (lipoxygenase homology) domain, which contains amino acids 2 to 119 in both proteins, and (b) the lipoxygenase domain, which ranges from amino acid 120 to 701 in ALOX12B and from 120 to 711 in ALOXE3. The PLAT/LH2 domain forms a β-sandwich containing two β-sheets and four β-strands. The function of the PLAT/LH2 domain is to mediate interaction with lipids or membrane-bound proteins [14]. ALOX12B and ALOXE3 belong to the same metabolic lipoxygenase pathway, where the product of ALOX12B is the substrate for ALOXE3.

*ALOX12B* and *ALOXE3* were firstly described as causative genes for ARCI by Jobard et al. [3]. Patients with mutations in *ALOX12B* or *ALOXE3* usually present with a mild phenotype including fine scaling and mild erythema. However, some patients may present with very severe erythroderma (personal communication from E.B.). In some patients, mutations in these genes lead to SICI. In most cases of SICI, mutations in *ALOX12B* were found, followed by mutations in *ALOXE3* [1,15].

To date, 88 pathogenic mutations in *ALOX12B* and 27 pathogenic mutations in *ALOXE3* have been reported in ARCI (Human Gene Mutation Database Professional 2020.3). In the present study, we present a large cohort of 224 patients affected with ARCI carrying mutations in *ALOX12B* or *ALOXE3*. We add 74 novel mutations in *ALOX12B* and 25 novel mutations in *ALOXE3*. Furthermore, we investigate the quality and spectrum of mutations, the potential hotspot regions, and the possible correlation of specific mutations with the phenotype.

## 2. Materials and Methods

In 224 patients with ARCI, mutations in *ALOX12B* or *ALOXE3* were detected using different sequencing methods, including Sanger sequencing and next-generation sequencing (NGS). Here, we present the results of the mutation testing for only one affected individual from each family. Our data show there were 224 patients of the cohort belonging to 223 families, as one patient was homozygous for both *ALOX12B* and *ALOXE3* mutations. This study was conducted according to the Declaration of Helsinki principles.

In all patients, genomic DNA was isolated from peripheral blood lymphocytes, and we performed PCR amplification using Sanger sequencing or NGS methods. All coding exons and flanking intronic sequences of *ALOX12B* (reference NM_001139.2, GRCh37.p13) and *ALOXE3* reference (NM_021628.2, GRCh37.p13) were analyzed. In general, Sanger sequencing methods for individual genes were used until 2010–2015, depending on the laboratory; NGS methods through multigene panel testing, either in a targeted way of through whole-exome sequencing, were applied and mutations validated by Sanger sequencing. In a large part of the cohort, DNA sequences were enriched by a HaloPlex Custom Kit or SureSelect Custom Kit (Agilent Technologies, Inc. Santa Clara, CA, USA). Resulting data were analyzed using an in-house bioinformatics pipeline and the commercial software SeqNext (JSI medical systems, Ettenheim, Germany).

For in silico analysis we used the following bioinformatics tools: Mutation Taster (http://www.mutationtaster.org/) [16], PolyPhen-2 (http://genetics.bwh.harvard.edu/pph2/) [17], fathmm v2.3 (http://fathmm.biocompute.org.uk/) [18], SIFT (http://sift.jcvi.org/) [19], Provean v1.1.3 (http://provean.jcvi.org/index.php) [20], NetGene2 v2.4 (http://www.cbs.dtu.dk/services/NetGene2/) [21], NNSplice version 0.9 (http://www.fruitfly.org/) [22], and SSP v2.1 (https://varseak.bio/, developed by JSI medical systems GmbH, Ettenheim, Germany).

In addition, the following databases were used: the Genome Aggregation Database version v2.1.1 (gnomAD; http://gnomad.broadinstitute.org/), HGMD^®^ Professional version 2020.3 (http://www.biobase-international.com/product/hgmd), Database of Single-Nucleotide Polymorphisms version build 151 (dbSNP; http://www.ncbi.nlm.nih.gov/projects/SNP/), PubMed (http://www.ncbi.nlm.nih.gov/pubmed/), and ClinVar version December 2020 (https://www.ncbi.nlm.nih.gov/clinvar/).

Alignments were retrieved from Ensembl 102 using the Eutheria Gen Tree node. Analysis and visualization was performed with Jalview version 2.11.1.3-j1.8 (https://www.jalview.org/).

## 3. Results

A genetic analysis of patients with ichthyosis over a period of 26 years revealed 170 families with mutations in *ALOX12B* or *ALOXE3* in our laboratories; a further 54 families were contributed by other coauthors within the ERN-Skin network. All patients are listed in Appendix A. A total of 31 of these pedigrees have already been published [1,3,23,24]. We found an additional 74 novel mutations in *ALOX12B* and 25 novel mutations in *ALOXE3*. All novel mutations are summarized in Table 1. To our knowledge, these mutations have not been reported in the literature before. Known and novel mutations were verified by HGMD^®^ Professional 2020.3. The results of prediction tools and databases for all novel mutations in *ALOX12B* and *ALOXE3* were listed in Appendix A. Our study increases the total number of known *ALOX12B* mutations to 162 and of *ALOXE3* mutations to 52.

### 3.1. Phenotype

Due to the high number of cases, this study did not aim to evaluate every single genotype–phenotype correlation. We obtained clinical data from almost two-thirds of our cohort. In many of these patients, we have only little information about the phenotype. We can confirm, however, that a collodion phenotype at birth, including the self-healing form, frequently occurs in patients with mutations in *ALOX12B* or *ALOXE3*, since a large proportion of our cohort shows these phenotypes. In our patients with available clinical data, about 76% of the patients with *ALOX12B* mutations were born with a collodion membrane (Appendix A). In *ALOXE3*, the proportion was about 36% (Appendix A). We did not find any association between specific mutations and specific clinical findings. The predominant clinical findings of most patients support the previous view of a mild phenotype including fine scaling and mild erythema in patients with *ALOX12B* and *ALOXE3* mutations. More severe erythroderma phenotypes occur occasionally but less frequently than milder phenotypes [1,3]. Some patients of our cohort are shown in Figure 1.

### 3.2. Spectrum of Mutations in ALOX12B

The most common type of mutations in *ALOX12B* in our cohort and in the literature are missense mutations (103 out of 161, corresponding to 64% of all cases), followed by frameshift mutations (32 out of 161, 20%), splice site mutations (12 out of 161, 7%), and nonsense mutations (10 out of 161, 6%). Less common types of mutations were a large deletion (1 out of 161), in frame deletions/insertions (2 out of 161), and a predicted start loss mutation (1 out of 161), which together represent 3% (Figure 2C).

We analyzed the distribution of the mutations within *ALOX12B*. We found that missense mutations occur in both the PLAT/LH2 and the lipoxygenase domain. In the beginning of the lipoxygenase domain from amino acid 120 to 251 (corresponding to exons 3 to 6), only 7 mutations were found, which is a strikingly smaller number of missense mutations compared with the rest of the lipoxygenase domain, where significantly more mutations are present (Figure 2A).

A total of 70 of 103 missense mutations (corresponding to 67%) were found in the second half of the protein from exon 9 to 15 (corresponding to amino acids 358 to 701) (Appendix A). Nonsense, frameshift, and splice site mutations seem to be evenly distributed throughout *ALOX12B*. Examination of the conservation of the amino acids in humans compared to other mammals revealed fewer conserved regions in the PLAT/LH2 domain and in the initial part of the lipoxygenase domain, including exons 3 to 6.

The large number of cases in this meta-analysis allows for an overview of the distribution of mutated alleles in affected individuals (Figure 2B). Patients carry these alleles either in a homozygous state or in a compound heterozygous state combined with another mutation in the same gene. The most frequent mutation by far is p.(Tyr521Cys) (61 out of 282 alleles, corresponding to 22%) followed by p.(Ala597Glu) (12 alleles, 4%) and p.(Val527Met) (8 alleles, 3%). About 2% of all alleles are the mutations p.(Ser16Leu) (7 alleles), c.1654+3A>G (7 alleles), p.(Arg432*) (5 alleles), and p.(Arg488His) (5 alleles). Mutations that occur even more rarely (1 to 4 alleles) represent 63% of all alleles.

### 3.3. Spectrum of Mutations in ALOXE3

In *ALOXE3*, most frequent mutations are nonsense and frameshift mutations (12 and 8, respectively), which lead to a predicted premature stop codon and potentially to nonsense mediated decay (combined, these are 20 out of 52 from our cohort and the literature, corresponding to 38%), followed by missense mutations (18 out of 52, 35%) and splice site mutations (10 out of 52, 19%). Three large deletions and one in frame deletion/insertion mutation were found (4 out of 52, 8%) (Figure 2F).

Compared to *ALOX12B*, there is no apparent uneven distribution of the mutations within the gene (Appendix A). Only 18 missense mutations were found, most of them are located in the lipoxygenase domain. It is possible that a larger dataset would reveal a similar pattern as in ALOX12B. Alignment analysis revealed highly conserved regions at the C-terminal end of the protein from amino acid 579. Compared to *ALOX12B*, *ALOXE3* has significantly fewer highly conserved regions.

Similar to *ALOX12B*, we investigated the frequency of mutated alleles in *ALOXE3*. The most frequent mutations is p.(Pro630Leu) (67 out of 166 alleles, corresponding to 40%), followed by p.(Arg234*) (35 alleles, 21%). Mutations that occur considerably less often are p.(Arg211*), p.(Glu319=), p.(Arg396Ser), and c.1393-1G>A (each 4 alleles, 2.5%). Mutations that occur in only 1 to 2 alleles correspond to 29% of all alleles (Figure 2E).

## 4. Discussion

An analysis of the mutational spectrum of *ALOXE3* in our cohort revealed the most recurrent mutations, p.(Pro630Leu) (40% of all alleles) and p.(Arg234*) (21% of all alleles). Our large cohort strongly supports the findings of Eckl et al. [25] and Fischer [11], who described these mutations as hotspot mutations. It is noticeable that both mutations account for almost two-thirds of all mutated alleles in our cohort. In *ALOX12B* the mutation p.(Tyr521Cys) occurs in 22% of all mutated alleles, and thus it occurs significantly more often than other mutations in this gene. Due to the different ethnic backgrounds of our patients carrying these mutations, a founder effect can be excluded. Patients from our cohort carrying *ALOX12B* mutations came from at least 21 different European and non-European countries. The mutation p.(Tyr521Cys) was found in patients from 15 different countries. In *ALOXE3*, mutations were detected in patients from at least 15 European and non-European countries. The mutations p.(Pro630Leu) and p.(Arg234*) were found in patients from 11 and 8 different countries, respectively (Appendix A). Some DNA sequences contain fragile sites that are more prone to mutation. These sites are often associated with specific dinucleotide or trinucleotide repeats or different methylation patterns. Furthermore, specific DNA structures can make DNA sequences more vulnerable to alteration. The DNA repair system can also be weakened in some sequence regions [26]. The exact mechanism for the occurrence of these hotspot positions in *ALOX12B* and *ALOXE3* is still unknown. Further investigations are needed to identify the underlying mechanisms.

The proportion of the types of mutations differs between *ALOX12B* and *ALOXE3*. In *ALOX12B*, 64% of all known mutations are missense mutations, followed by frameshift mutations (20%), splice site mutations (7%), and nonsense mutations (6%). As the hotspot mutations in *ALOX12B* are missense mutations, the proportion of missense mutations in patients is significantly larger than 64%. In *ALOXE3*, however, the proportion of missense mutations of all 52 known mutations amounts to 35%, followed by nonsense mutations (23%), splice site mutations (19%), and frameshift mutations (15%). Since the second most common hotspot mutation is a nonsense mutation, the proportion of mutations with a predicted premature stop codon in patients is very high. It is not yet clear how these differences between *ALOX12B* and *ALOXE3* can be explained.

This meta-analysis allows us to better understand the distribution of the mutations in *ALOX12B* and *ALOXE3*. In *ALOX12B*, 67% of all mutations were detected in the second part of the protein from exon 9 (from amino acid 358). At the beginning of the lipoxygenase domain from exon 3 to 6 (corresponding to amino acids 118 to 251), very few mutations were found. The cause of the uneven distribution of mutations within the gene is not yet clear. Compared to gnomAD, missense variants in this database are equally distributed within the gene (Appendix A). We speculate that some regions are much better conserved because they have important functional meaning. Less well-conserved regions may allow for changes that do not cause disease. Given the similar structure of ALOX12B and ALOXE3, it is surprising that we did not observe an uneven distribution in *ALOXE3* as we did in *ALOX12B*. However, the number of mutations in *ALOXE3* is still relatively low compared to *ALOX12B*, and therefore a similar pattern might emerge with increasing numbers of mutations to be identified in the future.

Analysis of the conservation of the amino acids in both proteins reveals highly conserved regions in a large part of *ALOX12B*, while only few regions are less conserved. Our results indicate a correlation between an increased occurrence of mutations and highly conserved regions. This can be explained by the fact that variants occurring at evolutionarily conserved sites are generally more deleterious compared to variants at nonconserved positions. However, *ALOXE3* has significantly fewer highly conserved regions compared to *ALOX12B*. Highly conserved regions can be found in the lipoxygenase domain, particularly at the end of the protein. Our results show that mutations in *ALOX12B* can be found significantly more often than in *ALOXE3*. It can be argued that the stronger conservation of amino acids in ALOX12B can explain the higher incidence of mutations in *ALOX12B*.

We obtained clinical data from almost two-thirds of our cohort, with a partial containing little information about the phenotype. We did not, however, find any indication for the correlation of specific mutations with phenotypes. Patients in our cohort were often born with a collodion membrane. In patients with obtained clinical data, a collodion phenotype occurred more often in patients with *ALOX12B* mutations (76%) compared to *ALOXE3* mutations (36%). This is in accordance with the findings of Simpson et al. [27], who found a collodion phenotype at birth in 71% of the *ALOX12B* cases and in 45% of the *ALOXE3* cases. Furthermore, we found a SICI phenotype in both *ALOX12B* and *ALOXE3* cases; this confirms the findings of Harting et al. [15] and Vahlquist et al. [1], who described mutations in *TGM1*, *ALOX12B*, and *ALOXE3* as a common cause of SICI.

Our study strongly expands the mutational spectrum in *ALOX12B* and *ALOXE3* and gives insights into the distribution of mutations within the genes and the occurrence and frequency of hotspot mutations. Further analyses can investigate the molecular genetic causes for the development of hotspot regions in these genes, function of the proteins and domains, and possible genotype–phenotype interactions.

## Figures and Tables

**Figure 1 genes-12-00080-f001:**
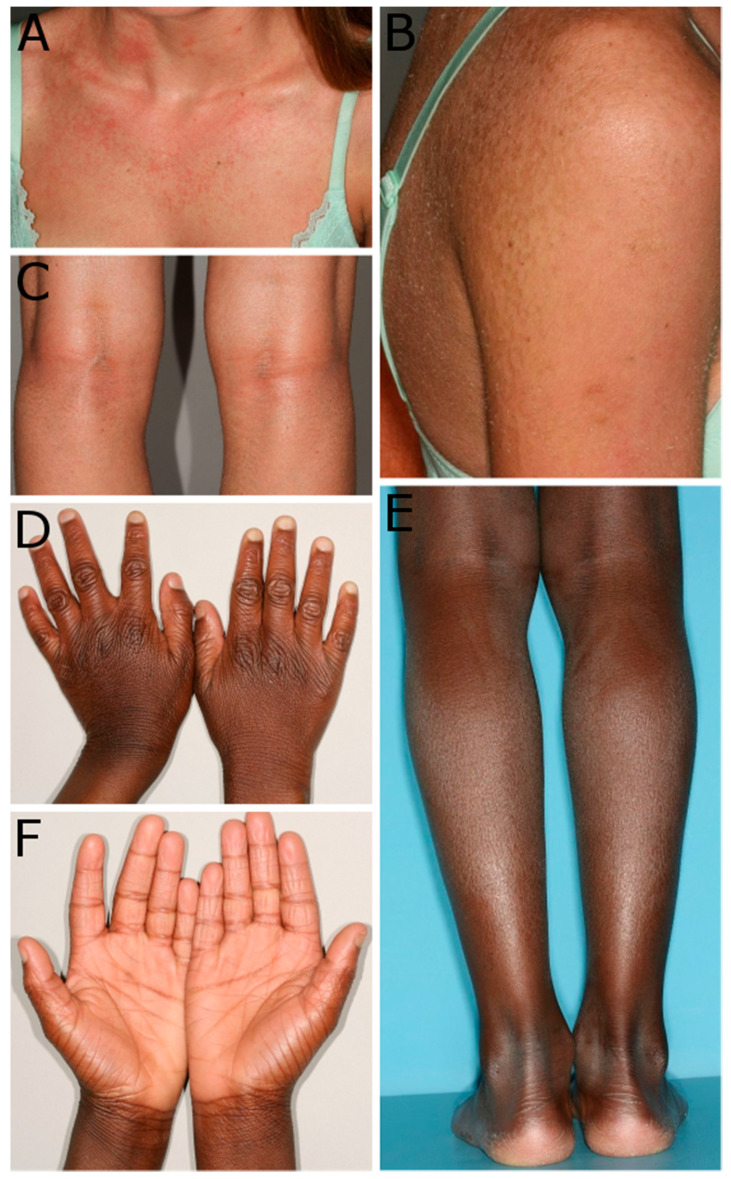
Examples of phenotypes of patients carrying mutations in *ALOX12B* and *ALOXE3*. (**A**–**C**): Patient 72 carries the mutations p.(Arg114Trp) and p.(Tyr383Met) in *ALOX12B* and shows an erythrodermic skin on the chest (**A**), scales on the arm and the back (**B**), and small scales on the popliteal area (**C**). (**D**–**F**): Patient 29 carries the homozygous mutation p.(Glu91*) in *ALOXE3* and presented with mild general ichthyosis. Patient 29 shows ichthyosis with knuckle pads on the dorsal sides of the hands (**D**), fine white scales on the legs (**E**), and hyperlinearity of the palmar sites of the hands (**F**).

**Figure 2 genes-12-00080-f002:**
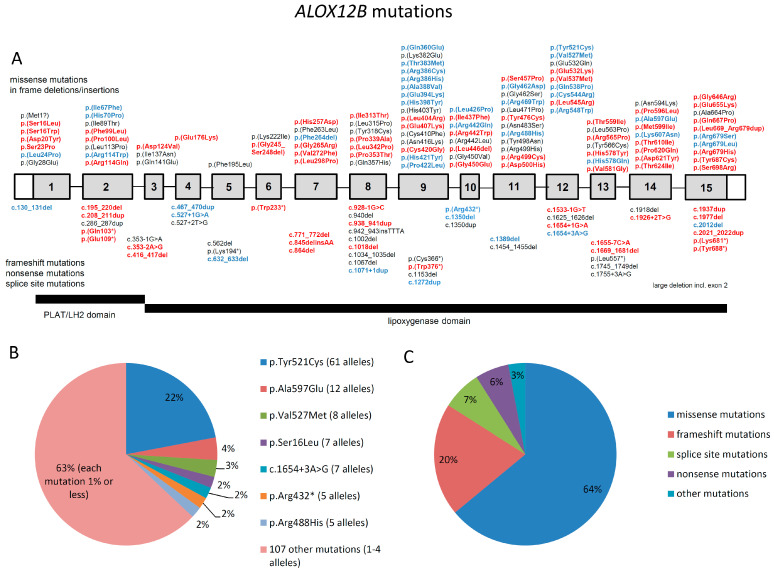
(**A**,**D**): Schematic presentation of the distribution of mutations in *ALOX12B* and *ALOXE3*. Red bold: novel mutations. Blue bold: known mutations, found in our cohort. Black, non-bold: known mutations, not found in our cohort. (**B**,**E**): frequency of mutated alleles in *ALOX12B* and *ALOXE3*. (**C**,**F**): frequency of different types of mutations in *ALOX12B* and *ALOXE3*.

**Table 1 genes-12-00080-t001:** Novel mutations in *ALOX12B* and *ALOXE3*.

*ALOX12B*				*ALOXE3*	
DNA Level	Protein Level	DNA Level	Protein Level	DNA Level	Protein Level
c.47C>G	p.(Ser16Trp)	c.1324C>T	p.(Arg442Trp)	c.57_63del	p.(Asp20Serfs*17)
c.47C>T	p.(Ser16Leu)	c.1336_1338del	p.(Leu446del)	c.271G>T	p.(Glu91*)
c.58G>T	p.(Asp20Tyr)	c.1349G>A	p.(Gly450Glu)	c.308A>C	p.(Gln103Pro)
c.67T>C	p.(Ser23Pro)	c.1369T>C	p.(Ser457Pro)	c.397A>G	p.(Arg133Gly)
c.195_220del	p.(lle66Argfs*50)	c.1427A>G	p.(Tyr476Cys)	c.680+1G>A	p.? splice site
c.208_211dup	p.(Lys71Thrfs*55)	c.1495C>T	p.(Arg499Cys)	c.758del	p.(Phe253Serfs*27)
c.297C>A	p.(Phe99Leu)	c.1498G>C	p.(Asp500His)	c.833A>C	p.(Tyr278Ser)
c.299C>T	p.(Pro100Leu)	c.1533-1G>T	p.? (splice site)	c.923T>C	p.(Leu308Pro)
c.307C>T	p.(Gln103*)	c.1594G>A	p.(Glu532Lys)	c.952dup	p.(Leu318Profs*58)
c.325G>T	p.(Glu109*)	c.1609G>A	p.(Val537Met)	c.957G>A	p.(Glu319=) splice site
c.341G>A	p.(Arg114Gln)	c.1634T>G	p.(Leu545Arg)	c.1031A>C	p.(Gln344Pro)
c.353-2A>G	p.? splice site	c.1654+1G>A	p.? splice site	c.1061G>A	p.(Trp354*)
c.371A>T	p.(Asp124Val)	c.1655-7C>A	p.? splice site	c.1164G>T	p.(Trp388Cys)
c.416_417del	p.(Ala139Glufs*37)	c.1669_1681del	p.(Arg558Serfs*2)	c.1193C>T	p.(Ser398Phe)
c.526G>A	p.(Glu176Lys)	c.1676C>T	p.(Thr559Ile)	c.1202T>A	p.(Leu401Gln)
c.698G>A	p.(Trp233*)	c.1694G>C	p.(Arg565Pro)	c.1246T>C	p.(Cys416Arg)
c.734_745del	p.(Gly245_Ser248del)	c.1732C>T	p.(His578Tyr)	c.1292dup	p.(His431Glnfs*90)
c.769C>G	p.(His257Asp)	c.1742T>G	p.(Val581Gly)	c.1393-1G>A	p.? splice site
c.771_772del	p.(His257Glnfs*116)	c.1787C>T	p.(Pro596Leu)	c.1786-2A>G	p.? splice site
c.793G>A	p.(Gly265Arg)	c.1797G>T	p.(Met599Ile)	c.1786-63_1807del	p.? gross deletion
c.814G>T	p.(Val272Phe)	c.1829C>T	p.(Thr610Ile)	c.1804dup	p.(Met602Asnfs*30)
c.845delinsAA	p.(Arg282Glnfs*92)	c.1859C>A	p.(Pro620Gln)	c.1812T>A	p.(Asn604Lys)
c.864del	p.(Val289Serfs*63)	c.1861G>T	p.(Asp621Tyr)	c.1937_1944del	p.(Ser646Thrfs*13)
c.893T>C	p.(Leu298Pro)	c.1871C>T	p.(Thr624Ile)	c.1954C>T	p.(Gln652*)
c.928-1G>C	p.? splice site	c.1926+2T>G	p.? splice site	deletion exon 15	p.? large deletion
c.938T>C	p.(Ile313Thr)	c.1936G>A	p.(Gly646Arg)		
c.938_941dup	p.(Ala316Profs*59)	c.1937dup	p.(His647Thrfs*50)		
c.1015C>G	p.(Pro339Ala)	c.1963G>A	p.(Glu655Lys)		
c.1018del	p.(Leu340Serfs*12)	c.1977del	p.(Arg660Glyfs*3)		
c.1025T>C	p.(Leu342Pro)	c.2000A>C	p.(Gln667Pro)		
c.1057C>A	p.(Pro353Thr)	c.2005_2037dup	p.(Leu669_Arg679dup)		
c.1071+1G>C	p.? splice site	c.2021_2022dup	p.(Asp675Thrfs*21)		
c.1127G>A	p.(Trp376*)	c.2036G>A	p.(Arg679His)		
c.1211T>G	p.(Leu404Arg)	c.2041A>T	p.(Lys681*)		
c.1219G>A	p.(Glu407Lys)	c.2060A>G	p.(Tyr687Cys)		
c.1258T>G	p.(Cys420Gly)	c.2064C>G	p.(Tyr688*)		
c.1309A>T	p.(Ile437Phe)	c.2094C>A	p.(Ser698Arg)		

## Data Availability

Data is contained within the article or Appendix A.

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
