# Peer review of "Meta-Analysis of Mutations in ALOX12B or ALOXE3 Identified in a Large Cohort of 224 Patients"

_genes, 2021, doi:10.3390/genes12010080_

Round 1
Reviewer 1 Report
Hotz et al. present an interesting meta-analysis on the mutational spectrum of the genes ALOX12B and ALOXE3. The manuscript is very well written and has a very high educational and informative value.
I suggest some minor changes, as specified below:
1- Figure 2 A: some mutations of the ALOX12B and ALOXE3 genes written in blue are unreadable.
2- Figure 2 C and F (frequency of different types of mutations in ALOX12B and
ALOXE3): it would be advisable to use the same color for each type of mutation. -> I mean, for example purple for nonsense mutations, orange for frameshift mutations and so on...
3- Page 10: "Due to the different ethnic backgrounds of our
patients carrying these mutations, a founder effect can be excluded." It would be interesting to summarize the geographical origin of the carriers of these two most recurrent mutations.
Author Response
Dear Reviewer,
Thank you very much for your remarks.
- We have improved the quality of figure 2 and increased the figure in a landscape format. The mutations should be more readable now.
- We have changed the colors in figure 2. Now each type of mutation has the same color.
- We have added some sentences in the discussion regarding the different ethnical backgrounds.
Reviewer 2 Report
Hotz et al herein describe and characterize a group of 224 genetically characterized patients, comprising a panel of novel detected pathological mutations in ALOX12B and ALOXE3 that are associated with autosomal recessive congenital ichthyoses, a group of disorders in cornification with generally mild erythrodermic symptoms. Previous mutations in these genes, as well as several other genes, had previously been described as causal for these conditions, but the authors broaden the mutational spectrum in this area.
The manuscript is clear, a good read, and presents all of the information that one would want from a descriptive report of novel mutations associated with a disease state or disease states. I have no major complaints or suggestions; the manuscript is of very high quality as is.
Author Response
Thank you very much for your comments.
Reviewer 3 Report
The manuscript focused to the mutational spectrum of the genes ALOX12B and ALOXE3, which encode the epidermal lipoxygenases arachidonate 12-lipoxygenase, 12R type (12R-LOX) and epidermis-type lipoxygenase-3 (eLOX3). Deficiency of 12R-LOX and eLOX3 enzymes disrupt the epidermal barrier function and the terminal differentiation of keratinocytes. The variants of the mutations may influence the autosomal recessive congenital ichthyosis (ARCI) phenotype including the self-improving collodion ichthyosis (SICI).
The manuscript strongly expands the mutational spectrum in ALOX12B and ALOXE3 (88 previously known and 73 novel pathogenic mutations in ALOX12B and 27 known and 25 novel pathogenic mutations in ALOXE3) and gives insights with meta-analysis into the distribution of the mutations).
The paper is well-written and emphasizes the importance of the novel knowledge of the ALOX12B and ALOXE3 pathology.
Based on detailed clinical data of this cohort, a further genotype-phenotype correlation study can be taken into consideration.
Author Response
Thank you very much for your comments.